# Optimizing Yield and Quality of Bio-Oil: A Comparative Study of *Acacia tortilis* and Pine Dust

**Gratitude Charis [1,\*], Gwiranai Danha [1] and Edison Muzenda [1,2]**

[1] Department of Chemical, Materials and Metallurgical Engineering, Botswana International University of Science and Technology, Palapye P Bag 016, Botswana; danhag@biust.ac.bw (G.D.); muzendae@biust.ac.bw (E.M.)

[2] Department of Chemical Engineering Technology, School of Mining, Metallurgy and Chemical Engineering, University of Johannesburg, P O Box, 17011, Doornfontein, Johannesburg 2094, South Africa

\* Correspondence: gratitude.charis@studentmail.biust.ac.bw; Tel.: +267-72-483-242

**Abstract:** We collected pine dust and *Acacia tortilis* samples from Zimbabwe and Botswana, respectively. We then pyrolyzed them in a bench-scale plant under varying conditions. This investigation aimed to determine an optimum temperature that will give result to maximum yield and quality of the bio-oil fraction. Our experimental results show that we obtain the maximum yield of the oil fraction at a pyrolysis temperature of 550 °C for the acacia and at 500 °C for the pine dust. Our results also show that we obtain an oil fraction with a heating value (HHV) of 36.807 MJ/kg using acacia as the feed material subject to a primary condenser temperature of 140 °C. Under the same pyrolysis temperature, we obtain an HHV value of 15.78 MJ/kg using pine dust as the raw material at a primary condenser temperature of 110 °C. The bio-oil fraction we obtain from *Acacia tortilis* at these condensation temperatures has an average pH value of 3.42 compared to that of 2.50 from pine dust. The specific gravity of the oil from *Acacia tortilis* is 1.09 compared to that of 1.00 from pine dust. We elucidated that pine dust has a higher bio-oil yield of 46.1% compared to 41.9% obtained for acacia. Although the heavy oils at condenser temperatures above 100 °C had good HHVs, the yields were low, ranging from 2.8% to 4.9% for acacia and 0.2% to 12.7% for pine dust. Our future work will entail efforts to improve the yield of the heavy oil fraction and scale up our results for trials on plant scale capacity.

**Keywords:** *Acacia tortilis*; biofuel; biomass; pine dust; pyrolysis

## 1. Introduction

The advanced biofuels market has been forecasted to expand at a compounded annual growth rate (CAGR) of 44% between the years 2017 and 2021. This trend is driven by higher demand for greener and cleaner energy, regional economic growth and more favorable trade balances through substitution of fossil fuel imports [1,2]. These aspirations have resulted in a spate of supportive policies from influential fuel traders like the United States and European Union, especially for advanced second-generation (2G) or third-generation (3G) biofuels that do not use edible feedstocks. Bio-oil is one such biofuel obtained from biomass pyrolysis, which has received much attention in research. The overarching goal is to find routes and processes that give higher yields and better quality products at lower costs or economic downstream upgrading methods. The option that is closer to the market is to obtain a moderately upgraded product that can substitute fuel oil for industrial applications, power generation and marine fuelling [3].

Lignocellulosic feedstocks are attractive for the production of such biofuels, due to their abundance and the possibility for higher energy recoveries since a larger fraction of the biomass is utilized compared

to the 1G counterparts, where only choice parts such as the grains are useful [4,5]. The most preferred lignocellulosic feedstocks are forestry/agricultural residues, or dedicated energy crops grown in marginal lands with no competition for food production. Charis et al. [6] reviewed the research and development (R&D) metrics of 2G biofuels compared to 1G for the period 2012–2017. They reported that 2G biofuels had 15% more researches and 23% more collaborations than 1G during the period, showing an increased interest in the former field.

## 1.1. Socio-Economic Background

Authors such as [4,7,8], have highlighted the need for Southern Africa to step up its participation in 2G biofuel research and to migrate from 1G biofuels that have limited feedstocks. Furthermore, using 1G feedstocks compromises food security directly, or indirectly through land-use change, which also results in increased greenhouse gas emissions. Southern Africa, in particular, has a wealth of lignocellulosic feedstocks in the form of agricultural waste, forestry residues and invasive species. However, the primary energy uses for biomass have been, mostly, the traditional open fire cooking methods. Since women and children are the most involved in such cooking activities, they are exposed to respiratory problems and endure arduous tasks such as gathering and chopping firewood [9,10]. Southern African nations are susceptible to energy poverty, especially in rural and remote areas, despite the high potential for renewable energy sources like solar, wind and biomass. The cases identified in two Southern African nations are examples of areas off the main electricity grid with a good inventory of biomass. The biomass identified in this research is encroacher bushes in Botswana and pine sawmill residues from Zimbabwe. In such cases, mini-grid power generation through the use of bio-oil in a stationary fuel oil generator could be an alternative to solar systems, especially when the pyrolysis system can be self-sufficient in terms of energy or can use a cheap, renewable energy source. Such a venture would be feasible if the bio-oil obtained does not need rigorous upgrading to match the counterpart fuel oils in terms of physical and fuel properties. Power generation is a priority application for energy impoverished Africa compared to the highly considered alternative use of bio-oil as a maritime fuel. Blending options with up to 25% pine bio-oil have proved to be miscible with heavy fuel oil (HFO) giving improved fuel flow properties while maintaining good engine properties [11]. If bio-oils with better fuel properties in terms of heating value can be obtained, blending ratios could be increased in favor of the bio-oil, and there could be a possibility of substituting the HFO entirely.

In this research, we assessed the conversion of the two types of forestry waste biomass to a 2G crude biofuel through pyrolysis and the subsequent potential application. Pyrolysis is a promising future source of fuels and chemicals that is resource-efficient, potentially auto-thermal, technically simpler, less capital intensive and operable at smaller scales compared to other thermochemical conversions. Pyrolysis can also be a good contributor to greening a brown economy by re-directing biomass feedstock that could otherwise have been burnt openly to be processed in a system largely regarded as carbon neutral. Moreover, its major products (bio-oil and char) both have high-value potential uses and are easier to handle than gas. Previous research by the same authors in [12,13] focused on the sources of lignocellulosic wastes, socio-economic and environmental drivers for their valorization.

## 1.2. Technical Background

Pyrolysis is a thermochemical conversion method in which a substance is heated in an oxygen-starved atmosphere to obtain solid, liquid and gaseous products. The yield of the various products depends on the process conditions, equipment used, feedstock types and pre-treatment regimes [14]. The liquid product (bio-oil), which is the primary target phase for this study, can be used as a fuel for heating, power and transportation after various levels of upgrading. It can also be a source of building blocks for the manufacture of various chemicals through polymerization of the monomer substrates. Bio-oil from pyrolysis also has its challenges that impede its uptake as a fuel including the high water content, high viscosity, low pH, instability, presence of solids, high oxygen content and low calorific value [15–17]. These properties explain the associated corrosive tendencies; ageing and phase

separation; immiscibility with hydrocarbon fuels; bad engine performance (due to injector blockages by solids and poor atomization) and high pumping costs [15].

### 1.3. The State of Research

In light of the challenges stated in Section 1.2, we review the progress in research on optimizing the quantity and quality of bio-oil obtained from pyrolysis in this section. We cover the intermediate to fast pyrolysis (IFP) range, which comprises moderate to high liquid yields (40–80 wt %) and hot vapor residence of 1–20 s. Most research within this scope has focused on optimizing pyrolysis process conditions and reactor designs for higher recoveries of bio-oil [15,18–21], while there has been limited research on conditions focusing on bio-oil quality improvement [14]. Temperature and heating rate have been identified as the most influential variables on bio-oil yields for a given biomass feedstock, among other factors like particle size and residence time. Consequently, the highest recoveries of bio-oil have been achieved with reactors that allow maximum heat transfer rate such as fluidized bed and free fall reactors, with temperatures and inert gas flow rates that facilitate fast pyrolysis [20]. Table 1 summarizes the operation philosophy of some of the most promising pyrolysis technologies and the status of their research, development or commercialization.

**Table 1.** Trending pyrolysis technologies. Adapted from [22,23].

| Pyrolysis Reactor | Description and Operation Philosophy | Operation Complexity and Max Oil Yield | Scale Up | Inert Gas Flow Rate | Particle Size | R&D Highest Status |
|---|---|---|---|---|---|---|
| Fixed bed | Biomass is placed immobile, above the inert gas distributor plate. Char remains in the reactor while oil and gas are collected downstream | Medium; up to 75 wt % oil | Hard | Low | Large | Few at pilot scale; Multiple lab scale |
| Bubbling fluidized bed (BFB) | Comprises reactor section with a continuous feed of biomass and high flow of inert gas to fluidize the particles. Char and sand are collected using cyclones. | Medium; up to 75 wt % oil | Easy | High | Small | Multiple demo and lab-scale plants |
| Circulating fluidized bed (CFB) | Similar to BFB, but collected char and sand are recycled through a combustor, which supplies hot sand to the fluidized bed. | High; up to 75 wt % oil | Hard | High | Medium | Multiple pilot and lab-scale plants |
| Ablative | Heat transfer to the biomass is direct from the walls of the reactor; no fluidizing gas. Biomass melts and vaporizes rapidly to form pyrolysis vapors. | High; up to 75 wt % oil | Hard | Low | Large | Few at pilot scale |
| Rotating cone | Heat transferred by reactor wall and hot sand, introduced into the rotating cone along with the biomass. The hot pyrolysis vapor is recovered from the bottom of the cone. | Medium; up to 70 wt % oil | Medium | Low | Medium | Demo/industrial scale |
| Screw/auger | Heat is mainly transferred by the wall surfaces. The biomass is moved along a heated cylindrical reaction zone by a screw | Low; up to 70 wt % oil | Easy | Low | Medium | Multiple pilot and lab-scale |

Amongst the in-process techniques used to improve the bio-oil quantity and quality, catalysis has enjoyed more attention than methods such as fractional condensation of condensable vapors and hot vapor filtration. Hot vapor filtration, which targets only the removal of ash, has been a big challenge with minimal research on long-term operations. This is due to complex requirements in terms of material specifications for the filter assembly and regeneration or cleaning mechanisms. Moreover, the trapped char catalytically cracks the pyrolysis vapors, reducing bio-oil yields by up to 20%. Simple in-process catalysis has been employed successfully using catalysts such as selected metals and zeolites to improve bio-oil yields and quality [15].

Condensers have traditionally been used to recover the condensable off-gases from the pyrolyzed matter. However, it is only the fractionated condenser systems that enable the recovery of targeted products of defined qualities. Fractional condensation has always employed multiple condensers, set at different temperatures, to selectively optimize the recovery of high calorific value oil and fractions with a high composition of speciality chemicals like methanol. Since the first condenser is typically set above 80 °C, water and light carboxylic acids are not recovered here. Therefore, the collected oil has higher stability, lower water content and about 2%–3% of acids, compared to 10%–15% in unfractionated bio-oil [3,24]. Since this is done within the same pyrolysis step, using the same process heat, the method is less energy-intensive than trying to distil the bio-oil product. Molecular distillation would result in the thermal decomposition of the largely unstable compounds in the bio-oil unless done carefully, with slower ramp-up rates at some stages such as coking. Such slow ramp-up rates would, however, lead to a very high overall energy consumption [24].

Several researchers have reported on various pyrolysis operation scales with multistage fractional condensation systems to obtain liquid bio-products of targeted physical or thermal properties. Gooty in [3] modeled the effect of both pyrolysis and condenser temperatures for a three-staged condenser system from the fast pyrolysis of birch bark and Kraft lignin. Gooty's system was composed of two cyclonic condensers and a condenser cum electrostatic precipitator (ESP) at outlet vapor temperatures of 105 °C, 0 °C and 38–56 °C respectively. For birch bark, the author obtained a yield of 35% of the dry oil (<1% water) with a higher heating value (HHV) of 31 MJ/kg. Chen et al. in [25], utilized four condensers in series at coolant temperatures of 32–44 °C, 25–27 °C, 22–25 °C and 22–25 °C, respectively, followed by an ESP. Their drier oils were obtained in condensers 2–4 with a water composition of 7.45%–7.82%, HHV of 22.6–23.5 MJ/kg and total yield of 14.3%. Oasmas et al. [26] pyrolyzed red oak wood in a fluidized bed with five stages of bio-oil recovery at vapor temperatures of 102 °C, 129 °C (ESP 1), 65 °C, 77 °C (ESP 2) and 18 °C. Most of the researches done on this subject have covered the continuous, fast pyrolysis regime using reactors such as fluidized bed and auger, with a few around the intermediate regime and reactors such as the fixed bed. According to [24], very little research on fractional condensation has been conducted at the lab scale.

*1.4. The Purpose and Significance of This Research*

In this research, we mainly sought to generate a profile for *Acacia tortilis* as a pyrolysis feedstock as, to our best knowledge, no work has been reported on this species. We conducted a comparative study with pine dust, a well-researched species. The pine dust residues in that geographical context have, however, not been experimented on before using pyrolysis. We aimed to find the optimum temperature for the maximum overall oil yield of both feedstocks, then the optimum primary (first) condenser temperature that gave the best fuel quality of bio-oil in a bespoke three-stage condensation line. The properties we considered for the bio-oil were HHV, viscosity, pH and specific gravity (SG), which we compared to fuel oil properties to assess the bio-oil's potential to substitute fuel oil in power generation engines. We also evaluated the chemical composition of the bio-oils using scanning calorimetry to investigate the presence of significant concentrations of extractable speciality chemicals. The research provides essential elementary information about the fuel and yield potential of acacia in comparison with pine when process and condenser temperatures are manipulated using a basic fixed bed reactor.

## 2. Materials and Methods

### 2.1. Methods of Feedstock Preparation and Characterization

Adequate knowledge of the supply capacity and the suitability of the biomass properties for thermochemical conversions is imperative for bioenergy projects to be sustainable [12]. To this end, we conducted proximate, ultimate and calorimetric analyses to determine the biomass quality and suitability for conversion through pyrolysis. We firstly milled dry pine dust of <10% moisture and *A. tortilis* shrubs using a JF 2-D chopper cum hammer mill (Staalmeester, Hartbeesfontein, South Africa) with a 0.8 mm sieve. We then sieved the mill product to obtain particles below 250 μm for the characterization tests. We used the American Society of Testing Materials (ASTM) standards for the characterization tests. Detailed test procedures were provided by the authors [27] and the properties are reported in the results section.

We carried out proximate analysis using a Leco 701 thermogravimetric analyzer (Leco TGA, St Joseph, MI, USA), which was intrinsically set according to ASTM standard E 1131–03 for compositional analysis using thermogravimetry. We then determined the moisture content (MC), volatile matter (VM) and ash and fixed carbon (FC) compositions using the mass loss plot from the TGA.

We also carried out a CNHSO composition analysis using a Flash 2000 CHNS analyzer (ThermoFisher Scientific, Waltham, MA, USA). Oxygen (O) was found by the difference.

We determined the higher heating value (HHV) of the two lignocellulosics using a CAL2K-2 bomb calorimeter (Digital Data Systems, Randburg, South Africa) employing the ASTM D5468–02 standard. The calorimeter was calibrated with benzoic acid. The HHV was determined as the average of two runs.

### 2.2. Pyrolysis

Of the many variables tested for optimum yields of bio-oil, temperature has been identified as the most fundamental and influential parameter on bio-oil yields [20]. It was our initial focus on these previously unstudied biomasses, from both a geographical (pine residues) and species (*A. tortilis*) viewpoint. We also optimized the quality of bio-oil by varying the primary condenser temperatures in the pyrolysis system while maintaining constant values for the secondary condensers.

We firstly comminuted the feedstocks using a hammer-chopper mill with a 5 mm sieve. We then screened the biomass to obtain a size range between 1.70 and 5.00 mm for the pyrolysis runs. We placed a mass of 200–220 g of the biomass inside a bench-scale fixed bed reactor (Figure 1). The reactor shell comprised an outside cylinder 55 cm high with an external diameter of 25.9 cm and inside diameter of 10.3 cm. The reaction took place in a removable cuboid-shaped holder of dimensions L—7.2 cm, W—7.2 cm and H—35 cm. The space between the inside and outside diameter had a compartment for the heating coils, next to the inner wall, and another one for cladding to insulate the reactor. We weighed the container weight and mass of biomass before each new run, while the mass of char was determined after the run. We also measured the weights of the bio-oil receivers before each run so that the net masses of the oils and the char could be determined by subtracting the mass of container from the mass of the container + oil or char.

We used the heat traced standpipe as the primary condenser (ES01) and Liebig shell and tube type for the secondary condensers (ES02 and ES03). We then adjusted the set point for the primary condenser was to 125 °C, while the secondary condensers were both set at 25 °C.

Subsequently, we purged the whole system for 2–3 min using inert nitrogen gas. We switched on the heating system at the SCADA graphical user interface (GUI) and ramped it at 66 °C per minute until the 450 °C setpoint, then maintained this temperature until the end of the run. The SCADA system captured the volumetric oil yields, and gas yields while showing the real-time temperature variations at different points of the bench-scale plant system. We ran each experiment until there were no more bubbles in the scrubber. We then weighed the bio-oil and chars and calculated the overall solid and liquid yields from the initial mass of feed. We repeated the pyrolysis runs at 500 °C, 550 °C and 600 °C. The optimum temperature was regarded as the one at which the highest total bio-oil wt % yield was

obtained. We selected the temperature range based on literature, which catalogues optimum pyrolysis ranges of 480–520 °C [15] and 400–600 °C [20] for various lignocellulosics such as herbaceous grasses, filter cakes and woody biomass. Bridgwater [15] comments that woody lignocellulosics occupy the upper end of these ranges due to the higher composition of temperature recalcitrant lignin, explaining why the authors selected a range of 450–600 °C.

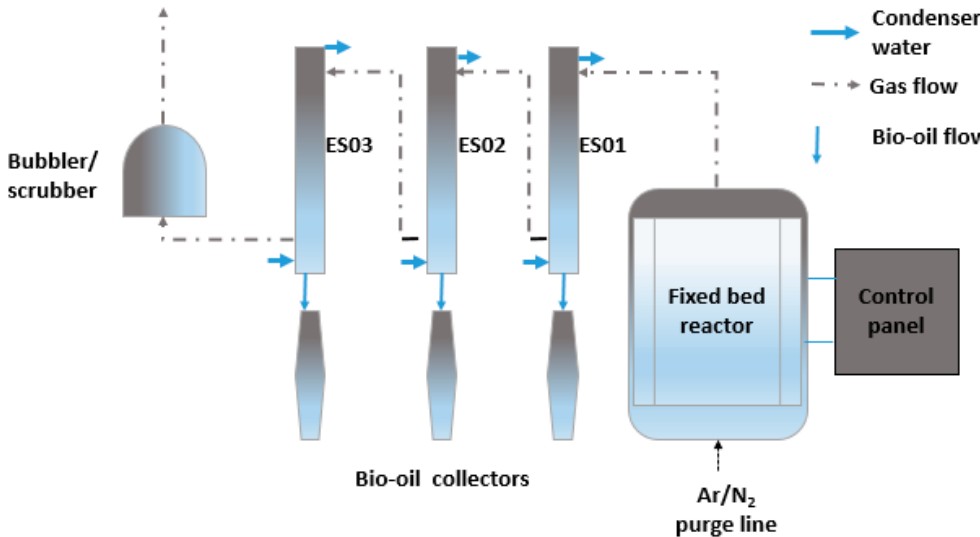

**Figure 1.** Schematic for the fixed bed pyrolysis unit.

To obtain the optimum primary condenser temperature, we ran the bench-scale plant at a pyrolysis temperature of 450 °C, while varying the primary condenser temperatures from 90 °C, 100 °C and then 110 °C, 125 °C and 140 °C [28]. Both secondary condensers were kept at 25 °C, which is below the dew point of water (50 °C) and some light acids so that these molecules would be collected at these points. We varied the primary condensation temperature above the 80 °C threshold recommended by [24] in such a way that only compounds of higher boiling and dew points with greater molecular weights would be retained successively at higher condenser temperatures. The highest condenser temperature of the exit hot vapor used, according to [24]'s review of multi-stage condensation systems was 135 °C, therefore, the authors went up to 140 °C. We initially regarded the optimum condenser temperature as the one that gave bio-oil of the highest specific gravity (SG), implying the least aqueous phase. However, the calorimetric tests carried out later on the oils proved to be a more decisive metric.

*2.3. Characterization of Products*

The product of interest was the bio-oil as a fuel for power generation and potentially, for vehicular applications. It was necessary to compare the properties of the bio-oil to diesel and fuel oils, hence the need for characterization. We also characterized both the primary and secondary condensates by gas chromatography (GC) Agilent 5975C (Agilent Technologies, Waldbronn, Germany) and a mass spectrometer (MS) (Agilent Technologies, Waldbronn, Germany) to assess if there were any useful extractable substances of significant concentrations. Finally, we characterized the residual biochar to assess its calorific fuel value, if it could be used to make charcoal briquettes.

The important physicochemical properties that we measured for comparison purposes include the SG, pH and viscosity. We obtained the SG by expressing the density of the bio-oil (mass/volume) as a ratio to the density of water and used it as an indicator of the water composition in the absence of a Karl Fischer titrator. We measured the pH was using a JENCO pH 6810 meter (Jenco instruments, San Diego, CA, USA) calibrated with pH 4.00, 6.86 and 9.18 solutions, while the viscosity was determined using a Thermo scientific Haake viscotester E (Waltham, MA, USA). We then measured the bio-oil thermal and fuel properties (represented by the calorific value) using an IKA C1 bomb calorimeter (IKA, Staufen,

Germany). Only the primary condenser samples could be tested as the secondary condenser bio-oils had a high aqueous composition that made it difficult for any combustion, even using at least 80 w/w% of a combustion aid. We poured each bio-oil sample was into a metal crucible then connected ignition wire and fuse combination before closing the bombshell. We then ran the enclosed system according to the DIN 51,900 standards and the calorific value was read off the panel. Where there was a difficulty in obtaining complete combustion, the test was repeated using a proportion of combustion aid of between 40% and 80%, whose calorific value was predetermined. We also tested the char calorific values using the same procedure. The actual HHVs of the char or bio-oil was then calculated using Equation (1).

$$HHV_{oil\ or\ char} = \frac{Total\ HHV - m.f\ of\ CA \times HHV_{CA}}{m.f\ of\ bio-oil} \tag{1}$$

where *CA* denotes the combustion aid, which was either paraffin or diesel; and m.f denotes mass fraction.

We assessed the chemical composition of the bio-oils using the GC–MS Agilent 5975C (Agilent Technologies, Waldbronn, Germany) equipped with a library. We initially filtered the samples using a syringe fitted with a 1 μm filter and a needle to eliminate small solid particles. We then dissolved them with 99.9% HPLC grade dimethyl ether, which had proved to be a better solvent compared to acetone. There are no standard tests and parameters yet for bio-oils set for engine applications, including procedures for GC–MS [29]. Therefore, we adopted this GC-MS test to resemble tests conducted on other bio-oils as well for fatty acid methyl ester (FAME) diesel [30–33]. We ran the samples in a GC–MS 5975C (Agilent Technologies, Waldbronn, Germany) using a DB-1HT 30 m × 250 μm diameter column with a 0.1 μm film. The flow rate of helium was set at 3 mL/min. We initially set the column temperature at 60 °C for 5 min, then, ramped at 5 °C/min to 235 °C and sustained this for 10 min. We then ramped it to 290 °C at 10 °C/min.

## 3. Results

This research brings together the results of overall pyrolysis optimization tests and selective optimization tests for the pine dust compared to the acacia. We had presented some characterizations of bio-oils from the selective optimization tests, namely the SG, pH and HHV, in [28]. However, we conducted further tests on the viscosity of the oils and their chemical composition and comprehensive comparative analysis of all results in this research to establish the feasible design and operation schemes for these residues.

We compared the properties of the bio-oil reported in this work to the properties of conventional types of diesel and fuel oils. The physicochemical properties of the bio-oil determine if the fuel could be used sustainably in unmodified form without affecting the engine through erosion or corrosion; or if they could be blended successfully. We also measured the HHV values of the chars to build a more holistic perspective on the socio-economic relevance of valorizing the wastes through pyrolysis.

### 3.1. Characterization of Feedstocks

Table 2 shows the results of the ultimate, proximate and calorimetric tests on the acacia and pine dust in comparison to other results for similar or related feedstocks.

**Table 2.** Characterization results: ultimate, proximate and higher heating value (HHV).

| Biomass | Ultimate Composition (%) | | | | | Proximate Composition (Dry Basis) (%) | | | | HHV (MJ/kg) |
|---|---|---|---|---|---|---|---|---|---|---|
| | C | H | N | S | O * | Ash | * FC | VM | MC | |
| *A. tortilis* | 41.47 | 5.15 | 1.23 | nd | 52.15 | 3.90 | 19.59 | 76.51 | 3.72 | 17.27 |
| Pine dust | 45.76 | 5.54 | 0.039 | nd | 48.66 | 0.83 | 20.00 | 79.16 | 6.50 | 17.57 |

nd—Not Detected; * by difference.

We could not find any literature on the characterization of *A. tortilis*; therefore, comparisons were made using other acacia species. We concluded that the results are comparable to those obtained in other literature, notably [34,35] where characterizations of *Acacia holosericea*, *Acacia mangium* and *Acacia auriculiformis* were discussed; and [36,37] where pine dust from two other nations was characterized. Both acacia and pine dust qualify as cleaner thermochemical feedstocks due to the good HHV, low N&S, acceptable ash content (below critical 6% for slagging and fouling) and high VM, which specifically favors higher pyrolysis oil yields. We had initially found the 'as received' pine dust to have an MC of 65.17%, which is higher than the recommended '<10%' for pyrolysis. Consequently, we sun-dried the biomass for at least 3 weeks to achieve an MC of 6.50%, below the '<10%' requirement. Due to the higher C and H content in pine and lower ash compared to acacia, one would expect pine to have a considerably higher HHVs; however, the acacia had a low MC, half that of pine. This could explain why the pine's HHV was not as high as expected. The pine dust from Zimbabwe had relatively lower N&S content compared the other pine forms, and the N was considerably lower compared to the acacia, showing that it was a cleaner thermochemical feedstock. Further discussions on these results will be made jointly with the results of pyrolysis and product characterization in Section 4.

*3.2. Optimum Pyrolysis Conditions*

3.2.1. Optimum Pyrolysis Temperature

Figures 2 and 3 show the variation of pyrolysis yields with temperature for both biomass forms. The maximum bio-oil yield (41.9 wt %) was obtained at 550 °C for the acacia, while pine dust had its maximum yield (46.1 wt %) at 500 °C. The optimum pyrolysis temperature for pine dust (500 °C) corresponded to that obtained for Indian pine and Canadian mixed sawdust where maximum yields of 39.37% and 45% were obtained in a fixed bed reactor [38,39]. Bridgwater [15] claims in his review of fast pyrolysis that optimum bio-oil yields from lignocellulosic biomass should be obtained between 480 and 520 °C, with grasses occupying the lower end and woody biomass the higher end.

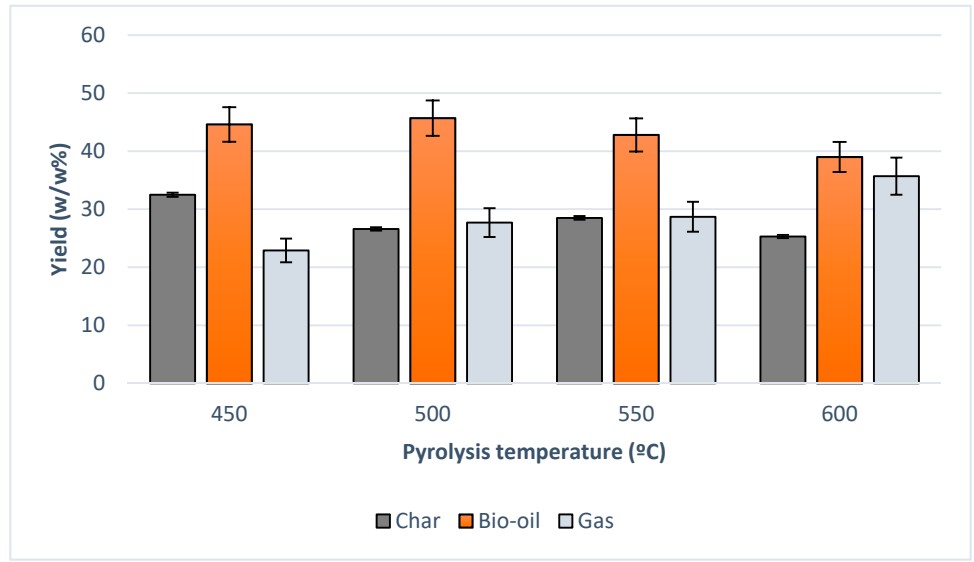

**Figure 2.** Variation of product yields with pyrolysis temperature for pine dust.

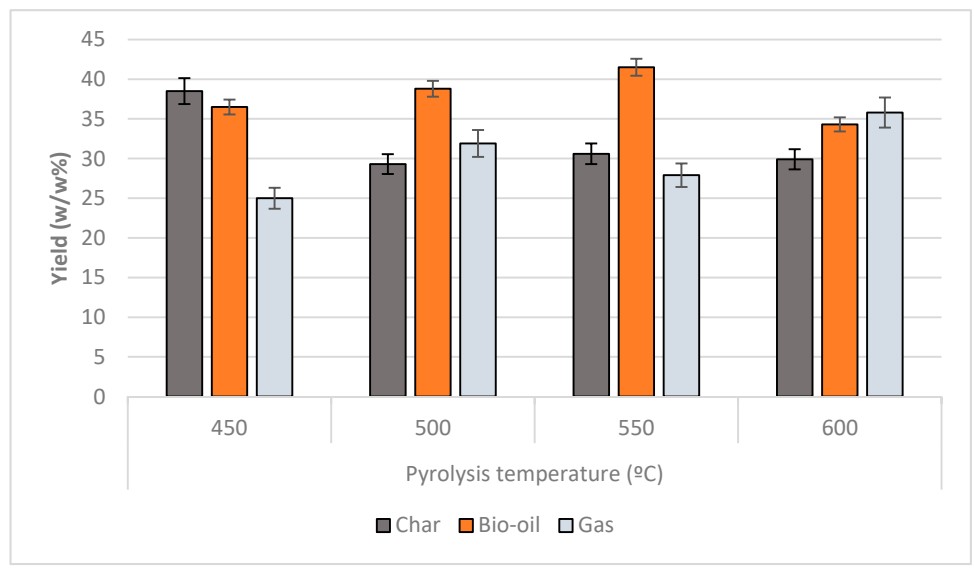

**Figure 3.** Variation of product yields with pyrolysis temperature for acacia.

With no research reported on *Acacia tortilis*, we could only compare the pyrolysis results from this research to the results obtained on other acacia species. Ahmed et al. [40] compared the bio-oil yields from the intermediate pyrolysis of various parts of *Acacia cincinnata* and *Acacia holosericea* in a fixed bed reactor at a temperature (500 °C. They obtained yields of 45.31–52.95 wt % for the *A. cincinnata* and 41.24–46.92 wt % with the *A. holosericea*, which are comparable to the maximum *Acacia tortilis* yield of 41.9 wt %. Reza et al. [34] used a fixed bed reactor to pyrolyze *A. holosericea* and experimented at two temperatures, 500 °C and 600 °C, obtaining a maximum bio-oil yield of 37.61 wt % at 600 °C. The optimum pyrolysis temperature of 550 °C for *Acacia tortilis* was therefore credible and higher than that of pine due to the higher temperature recalcitrance of the hardwood (acacia) compared to the softwood (pine).

As stated in Section 1.3, temperature and heating rate have been the most experimented variables deemed as the most influential to pyrolysis oil yields and the product distribution [20]. In this case, we only manipulated the temperature since it is difficult to maintain a constant local heating rate for large particle sizes and where several reactions occur. The overall pyrolysis temperature was, therefore, the distinguishing variable affecting the oil yield when equal ramp rates and biomass particle sizes were used. Raising the pyrolysis temperatures until the optimum point increases the heat transfer rates, which favor oil yields to higher localized heating rates. In the fluidized bed, conical spouted bed and circulating bed reactors the small particle sizes and fluidization effect cause even higher heating rates and higher oil yields.

On the other hand, ablative and rotating cone reactors use the high rate of heat transfer from the walls of the reactor, which are typically at elevated temperatures. These higher heating rates, according to Uzun et al. [41], accentuate the reduction of the water content in bio-oil explaining such fast pyrolysis conditions would give higher quality bio-oils compared to the intermediate regime in this research. Typically, such high heating rates should be coupled with moderately high inert gas flow rates to reduce the hot vapor residence time; otherwise, they crack or repolymerize, leading to low oil yields [20]. Such a continuous purge stream was another missing aspect of achieving a fast pyrolysis mode in the case of the fixed bed reactor where the inert gas was used to purge out *the* air at the initial stages of the experiment only.

The ensuing sections focus on the results we obtained from the attempt to optimize the quality of bio-oil by separating the heavier oil from the aqueous light phase using various condenser temperatures.

### 3.2.2. The Effect of Fractionation on pH

Figure 4 shows the pH of the separated fractions, the heavier oil or primary condensate and the lighter oil or secondary condensate.

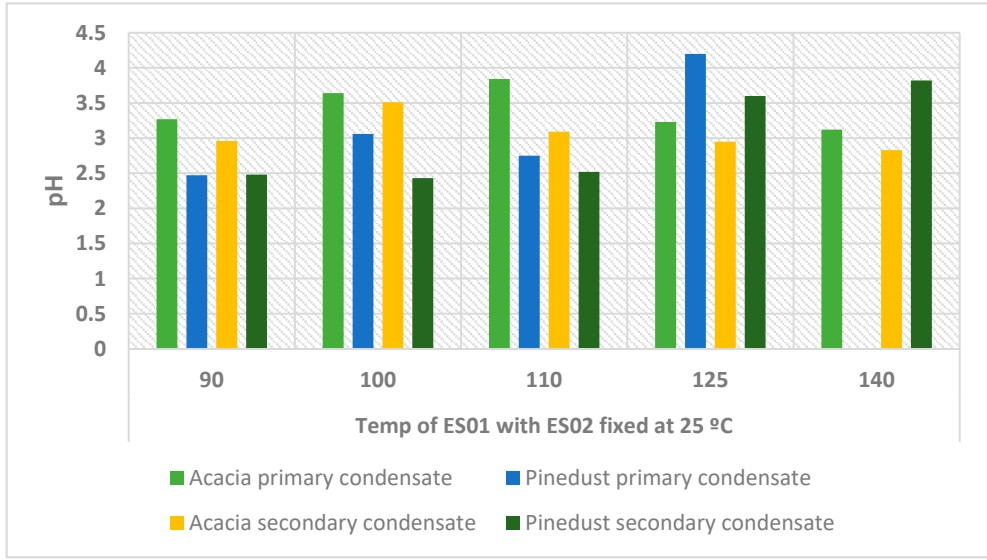

**Figure 4.** pH of various bio-oil fractions at different primary condenser temperatures for a sample.

The results indicate that the heavier oil fraction obtained in the primary condenser (ES01) has a higher pH, suggesting that most acids preferentially reported to the more aqueous secondary condensates. The *A. tortilis* primary condensates had a higher average pH (3.42) compared to pine dust (average 3.12), although the highest pH was obtained for the pine (4.20). These averages are however, higher and consequently better than the average of 2.5 estimated in the literature for woody bio-oils in general [15]. High acidity in the bio-oil makes it corrosive, especially with metallic vessels or pipes.

We also observed that there was a drastic reduction in the heavier oil yield for the pine at the primary condenser temperature of 140 °C. This could suggest that most of the molecular compounds in pine pyrolysis vapors have low dew points, vaporizing at higher condensation temperatures such as 140 °C and resulting in very low heavy oil yields. We could not conduct the pH, SG and HHV characterizations on the pine bio-oil obtained at this temperature due to the low yield of 0.2% (Table 3). This explains why there are gaps in Figures 4 and 5.

**Table 3.** Actual yield of primary condensate at various primary condenser temperatures.

| Temp of ES01 | | 90 °C | 100 °C | 110 °C | 125 °C | 140 °C |
|---|---|---|---|---|---|---|
| *Acacia tortilis* | Heavy oil yield | 6.0% ± 0.2% | 5.4% ± 0.1% | 4.9% ± 0.1% | 2.8% ± 0.1% | 3.9% ± 0.1% |
| | Total oil yield | 38.8% ± 1.0% | 41.9% ± 1.0% | 40.8% ± 1.0% | 36.5% ± 0.9% | 37.5% ± 1.0% |
| Pine dust | Heavy oil yield | 15.4% ± 0.4% | 10.7% ± 0.3% | 12.7% ± 0.3% | 7.2% ± 0.2% | 0.2% |
| | Total oil yield | 34.7% ± 0.9% | 46.1% ± 1.2% | 44.8% ± 1.1% | 44.4% ± 1.1% | 33.5% ± 0.9% |

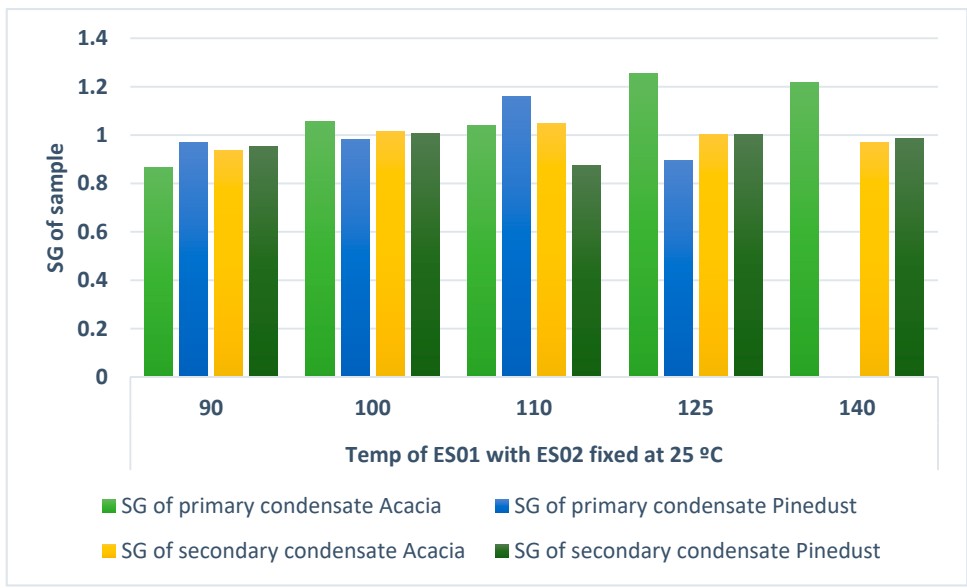

**Figure 5.** Specific gravity (SG) of various bio-oil fractions at different primary condenser temperatures.

### 3.2.3. The Effect of Fractionation Temperature on SG, Heavy Oil Yields and Quality

Figure 5 shows the variation of SG with primary condenser temperature, which was used as an indicator of the aqueous composition in the oil.

The higher the SG, the lower the aqueous content and the higher the perceived quality of the product. This is because pure pyrolysis oil has a higher SG than water; therefore, as the purity of oil increases, the SG increases too. Using this metric, the optimum quality bio-oil would be at 125 °C for acacia (SG of 1.257) and at 110 °C for pine (SG of 1.159). At primary condenser temperatures above the boiling point of water (100 °C), the quality of the oil increased, as expected, since more of the water reported to the secondary receivers. However, for pine, beyond the 110 °C we observed a reduction in the yield of the heavier oil (see Table 3) and the least total yield of bio-oil was obtained at 140 °C. One of the pine dust pyrolysis runs at 125 °C also had a zero yield, although it was not used in the calculation of the average. We suspect that the liquid that accumulated in the other two runs could be the result of back mixing of the vapors at that temperature, which caused the accumulation of an oil water mixture of a low SG. This could have been due to the design or orientation of the tubes at the condensers inlets and outlets. The smaller yield of the dry, heavier oil at higher temperatures could be due to lower dew points of molecular compounds in the pine bio-oil [24]. The 140 °C condenser temperature did not seem to have a major effect however on the acacia, most probably because of its higher temperature recalcitrance as a hardwood, therefore the total and heavier oil yields were not significantly reduced.

### 3.2.4. Properties of the Bio-Oils Obtained at Various Condenser temperatures

Physico-Chemical Properties of the Heavier (Choice) Bio-Oil

Since we target to use the bio-oil in a moderately upgraded form as a fuel for power generation, it is necessary to compare its properties to those of conventional fuels such as diesel, heavy and light fuel oil whose engines could be compatible with the bio-oil. These average HHV values are shown in Table 4, along with the measured values for the viscosity. We used the heavier bio-oil fraction properties in the comparison since they were more comparable with conventional fuels, especially the HHVs. We could not test the secondary condenser oils for HHV due to the high water composition that prevented combustion, even with combustive aids.

**Table 4.** Various bio-oil physicochemical properties compared to conventional fuels.

| | | Viscosity (mPa·s) at 25 °C | | HHV (MJ/kg) | | SG | |
|---|---|---|---|---|---|---|---|
| | 90 °C | *A. tortilis* oil | Pine dust oil | *A. tortilis* oil | Pine dust oil | *A. tortilis* oil | Pine dust oil |
| | | 2217.6 | 3043 | 4.310 | 5.227 | 0.867 | 0.968 |
| Primary condenser (ES01) Temp °C | 100 °C | 4804.8 | 13,951 | 21.412 | 9.235 | 1.057 | 0.982 |
| | 110 °C | Too little | 10,151 | 23.610 | 15.780 | 1.040 | 1.159 |
| | 125 °C | Too little | 2142 | 26.191 | 0.6338 | 1.257 | 0.894 |
| | 140 °C | Too little | Very little | 36.809 | Very little | 1.217 | Very little |
| Conventional diesel | | 2178 | | 43–45 | | 0.844 | |
| Light fuel oil (LFO) | | - | | 42–44 | | 0.85–0.910 | |
| Heavy fuel oil (HFO) | | >17,800 at 50 °C | | 40 | | 0.940–0.989 | |

The viscosities of the two bio-oils are on the higher end of the range reported in the literature for woody bio-oils, of 30–12,000 mPa·s or more [15]. For the best quality of pine bio-oil obtained at 110 °C, we obtained a viscosity of 10,151 mPa·s, which is four times the viscosity of diesel (2178 mPa·s) and almost half that of HFO (17,800 mPa·s) at 50 °C. The bio-oils can, therefore, readily substitute HFO in blends and even lower energy requirements for heating and pumping HFO. The blending ratio with pine bio-oil would, however, be limited given its low HHV compared to the acacia bio-oil [11]. Unfortunately, we could not measure the viscosities of acacia bio-oil obtained at higher temperatures due to their small volumes; however, if the trend against the pine dust values continued as expected, they would still be way below the HFO values. Meanwhile, the acacia bio-oil's HHV values also compare well to HFO, though the SG is about 1.2 times more, tallying with the average value of bio-oil density (1200 kg/m$^3$) provided by [42]. This means that the acacia heavy bio-oil at ES01 of 140 °C has 92% of the energy content in HFO on a weight basis, but has 113% on a volumetric basis. This bio-oil could, therefore, be a good substitute provided that it is obtained in reasonably large quantities to make the process economically justifiable. The 15.780 MJ/kg obtained for the pine dust bio-oil is slightly lower than the 17 MJ/kg reported in the literature and much less than the HHVs of diesel, LFO or HFO [15].

The Chemical Composition of the Bio-Oils

In this section, we have only presented the GC–MS results of best quality primary condensates and their corresponding secondary condensates as representative samples. We managed to identify 95 compounds from the primary condensate and 76 from the secondary condensates for the *A. tortilis* bio-oil at the primary condenser temperature of 140 °C. For the pine bio-oil at a temperature of 110 °C, 107 compounds were detected in the primary condensate, while only 87 were identified in the secondary condensate. Bio-oil is typically composed of more than 300 oxygenated compounds [17]. However, not all of the compounds can be easily identified by the conventional GC–MS due to their complex matrix in the bio-oil; therefore we recommend a GC × GC, with a higher resolution in 3D, to identify most of the compounds. Tables 5 and 6 show the eight compounds with the largest peak areas for the sampled primary and secondary condensates.

**Table 5.** GC–MS analysis for *Acacia tortilis* primary and secondary condensates obtained at optimal condition (primary condenser temp of 140 °C).

| | Compound | Compound Class | Molecular Weight (g/mol) | Area % | Retention Time (min) |
|---|---|---|---|---|---|
| Primary condensate | Cresol | Methylphenol | 138.16 | 7.759 | 8.23 |
| | Phenol, 4-ethyl-2-methoxy- | Phenol | 152.19 | 5.325 | 10.31 |
| | Phenol, 2,6-dimethoxy- | Phenol | 154.16 | 5.120 | 11.83 |
| | Phenol, 2-methoxy- | Phenol | 164.20 | 4.847 | 5.89 |
| | Benzene, 1,3-bis(1,1-dimethylethyl)- | Hydrocarbon | 190.33 | 4.390 | 10.09 |
| | Phenol, 2,4-bis(1,1-dimethylethyl)- | Phenol | 278.50 | 4.254 | 16.28 |
| | 5-tert-Butylpyrogallol | Phenol | 182.22 | 3.302 | 16.16 |
| | Heneicosane | Alkane hydrocarbon | 296.583 | 3.3148 | 18.57 |
| | **Others** | | | | |
| | Benzoic acid | Carboxylic acid | 122.12 | 0,6791 | 8.48 |
| | Cyclopenten-1-one, 2-hydroxy-3-methyl- | Ketone | 68.12 | 2,1704 | 4.73 |
| | Octane, 3-ethyl- | Alkyl hydrocarbon | 142.28 | 1.073 | 5.54 |
| | Furaldehyde phenylhydrazone OR Furfuraldehyde | Aldehyde | 96.0841 | 0.1729 | 19.53 |
| Secondary condensate | Phenol, 2-methoxy- | Phenol | 164.20 | 15.638 | 6.06 |
| | Phenol, 2,6-dimethoxy- | Phenol | 154.16 | 12.377 | 12.38 |
| | 2-Cyclopenten-1-one, 2-hydroxy-3-methyl- | Ketone | 112.13 | 7.533 | 5.01 |
| | Hydroquinone mono-trimethylsilyl ether | Phenol | 110.03 | 6.052 | 16.29 |
| | 4-Methoxy-2-methyl-1-(methylthio)benzene | Hydrocarbon (phenylpropanes) | 168.26 | 5.103 | 14.40 |
| | Benzene, 1,3-bis(1,1-dimethylethyl)- | Hydrocarbon | 190.32 | 3.960 | 10.12 |
| | 2-Cyclopenten-1-one, 3-ethyl-2-hydroxy- | Ketone | 126.15 | 3.764 | 6.83 |
| | Cyclohexanol, 2,2-dichloro-1-methyl- | Alcohol | 183.03 | 2.138 | 4.49 |
| | **Others** | | | | |
| | Trans 2-(2-Pentenyl)furan | Furan | 136.19 | 0.2457 | 7.36 |
| | 3,4-dimethylcyclohexanol | Alcohol | 128.21 | 0.3821 | 4.66 |

**Table 6.** GC–MS analysis for pine dust primary and secondary condensates obtained at optimal condition (primary condenser temp of 110 °C).

| | Compound | Compound Class | Molecular Weight (g/mol) | Area % | Retention Time (min) |
|---|---|---|---|---|---|
| **Primary condensate** | Phenol, 2-methoxy- | Phenol | 164.20 | 11.066 | 6.01 |
| | Cresol | Methylphenol | 138.16 | 7.154 | 8.26 |
| | Benzene, 1,3-bis(1,1-dimethylethyl)- | Hydrocarbons | 190.33 | 4.935 | 10.10 |
| | 1,2-Cyclopentanedione, 3-methyl- | Ketone | 183.07 | 4.851 | 4.77 |
| | Hexadecane | Hydrocarbon | | 4.1684 | 18.80 |
| | Phenol, 2,4-bis(1,1-dimethylethyl)- | Phenol | 278.5 | 4.047 | 16.27 |
| | Homovanillyl alcohol | Alcohol | 168.19 | 3.8349 | 16.04 |
| | Phenol, 4-ethyl-2-methoxy- | | 152.19 | 3.6806 | 10.32 |
| | **Others** | | | | |
| | trans-Isoeugenol | Phenol | 164.20 | 2.544 | 13.37 |
| | Tridecane, 7-hexyl- | Hydrocarbon | 268.5 | 1.6788 | 22.42 |
| | Propanal, 2-propenylhydrazone | Aromatic aldehyde | 126.15 | | |
| | Methoxyacetic acid, nonyl ester | Monocarboxylic acid and ether | 216.32 | 0.2737 | 7.23 |
| **Secondary condensate** | Phenol, 2-methoxy- | Phenol | 164.20 | 14.526 | 5.99 |
| | Cresol | Methylphenol | 138.16 | 14.093 | 8.38 |
| | Phenol, 4-ethyl-2-methoxy- | Phenol | 152.19 | 9.343 | 10.42 |
| | 2-Cyclopenten-1-one, 2-hydroxy-3-methyl- | Ketone | 112.13 | 7.509 | 4.93 |
| | Phenol, 2-methoxy-4-(1-propenyl)-, (Z)- | Phenol | 164.20 | 4.767 | 14.44 |
| | Phenol, 2-methoxy-4-propyl- | Phenol | 166.22 | 2.871 | 12.53 |
| | Benzene, 1,3-bis(1,1-dimethylethyl)- | Hydrocarbon | 190.32 | 2.281 | 10.10 |
| | Eugenol | Guaiacol (phenol) | 164.20 | 2.276 | 12.23 |
| | **Others** | | | | |
| | Propanoic acid, 2-methyl-, 2,2-dimethyl-1-(2-hydroxy-1-methylethyl)propyl ester | Carboxylic acid and ester | 244.37 | 0.7445 | 5.76 |
| | Maltol | Pyranones (ketones) | 126.11 | 1.47 | 6.47 |

For *A. tortilis*, we observed that cresol had the highest composition detected in the primary condensate, while it was next to Phenol, 2-methoxy- in the pine dust primary condensate. In the secondary condensates, Phenol, 2-methoxy and Phenol, 2, 6-dimethoxy were the most concentrated in the acacia, while Phenol, 2-methoxy- and cresol were the highest in the pine. The secondary condensates had characteristically high concentrations of specific compounds in both biomass forms and could be used as a source of chemicals since they are not useful for fuel. The phenols in the secondary condensers were of significant concentrations on a dry basis and could be extracted for use as food additives—mostly as flavorings, spices, seasonings and colorings—while cresol is an active ingredient in disinfectants [43]. We also identified other compounds in both bio-oils including acids like acetic acid, benzoic acid and homovanillic acid; esters such as succinic acid and the nonyl tetrahydrofurfuryl ester;

food flavorants and their precursors such as maltol and guaiacol (also used as a pesticide) and ketones such as acetone and other types of phenols. We did not manage to detect Levoglucosan as expected in such lignocellulosic biomass [44] probably because it was present in tiny quantities. However, we detected another type of anhydrous sugar compounds, 1,4:3,6-Dianhydro-.alpha.-d-glucopyranose, in the dry pine bio-oil with a peak area of 1.22%. Lyu et al. [45] also observed a shallow composition of anhydrosugars (<1 wt %) at a pyrolysis temperature of 480 °C, which increased at 500 °C and 520 °C, attaining a maximum of 4.1 wt % at 550 °C.

The GC–MS results that we obtained are comparable to those obtained by Ahmed et al. [40] for the trunk of *Acacia cincinnata* and *Acacia holosericea*, who reported that phenolic compounds and their derivatives were the major compounds detected in the bio-oil samples. For instance, the *Acacia holosericea* trunk was found to have the following sampled areas: phenol, 2,6-dimethoxy- (5.36%), cresol (3.39%), (E)-2,6-dimethoxy-4- phenol (3.35%), phenol, 2-methoxy- (3.23%), benzene, 1,2,3-trimethoxy-5-methyl- (3.22%), 2- cyclopenten-1-one, 2-hydroxy-3-methyl- (2.04%), 2-furancarboxaldehyde (2.99%), isoeugenol 2 (2.08%), guaiacol, 4-ethyl- (2.06%) and 2-methoxy-4-vinylphenol (1.72%) [40]. The high presence of phenolics in both the acacia and pine primary oils is typical as reported by Muda et al. [46], who recorded a high composition of phenolics in bio-oil obtained from the subcritical water treatment of oil palm trunk. The compound with the highest composition in their bio-oil was phenol with a peak area of 17.11%, while 2,6-dimethoxy-phenol and 4-ethyl-2-methoxy-phenol had 7.10% and 1.78% respectively. Lyu et al. [45] have also identified phenol peak areas of up to 20% for the pyrolysis of wood and rice straw, while Kim et al. [47] mention a composition of 0.1–3.8 wt %. In terms of peak areas, the findings in our research are, therefore in agreement with the literature cited. Kim et al. [47] also mentioned that the pyrolysis of softwood lignins yielded mainly guaiacols, including the eugenols, vanillin and homovanillin, which had a significant presence in the pine bio-oil. The composition of the acacia bio-oil that we observed was also consistent with the observation that hardwood lignins give both guaiacols and syringols, including phenols, methoxy phenols and cresol. The high composition of phenols that we observed could also be due to the low pyrolysis temperature of 450 °C used for the fractional condensation experiments, corroborating with research done by Lyu et al. [45]. In their research, the phenolic content in bio-oil increased from 3.9 to 5.5 wt % when the temperature was decreased from 550 to 480 °C.

The reported compositions of bio-oil differ widely in the literature depending on the biomass source and also on process conditions used [20]. At a higher condensation temperature for instance, in the case of acacia, the product was a tar-like oil of high HHV, rich in complex phenolics (48% peak area) with ~30% hydrocarbons and the rest of the area occupied by other compounds—on a dry basis. The dominance of complex phenolics and aromatics in such tar-like oil corroborates with literature such as Muda et al. [46] and these authors also obtained an exceptionally high HHV value for that bio-oil (33.2 MJ/kg). The characteristically high HHV of this tar-like oil, which is similar to that obtained in this research, seems to contradict with the general trend that the bio-oil with a higher percentage of phenolics could be unuseful as a fuel [20]. The high HHV could be due to the complex type of phenolics comprising methyl and methoxy groups and the significant amount of long-chain carboxylic acids, hydrocarbons and ketones (total 38% peak area) in this tar-like oil. Coupled with the low aqueous composition indicated by the high SG, it is logical that the primary condensate should have such a high calorific value. On the contrary, the secondary condensates had ~26% peak area occupied by hydrocarbons, ketones and carboxylic acids and a higher aqueous composition demonstrated by the low SG.

### 3.2.5. Uncertainty Analysis

We assessed the margin of error that could occur in the pyrolysis temperature range for the bio-oil, char and gas separately by obtaining four readings at a single temperature then calculating the relative standard error (%). This value of the error was then applied across the various temperatures. For instance, from Figure 3, the chars were liable to an error of 4.24% while the bio-oil and gas had

margins of 2.57% and 5.28% respectively. This same logic was applied for the variations in condensation temperature, which affected the oil yields in Table 2.

## 4. Overall Discussion

The characterization results had already indicated a higher VM in pine (79.16%) compared to acacia (76.5%), which explains why the total yield of oil from pine was higher for all cases than for acacia. The heavier oil yield was also generally higher for the pine dust oil compared to *A. tortilis* (see Table 3). For instance, the best quality pine dust heavy oil in terms of HHV, obtained at the primary condenser temperature of 110 °C, was 12.7% compared to 3.9% for the acacia at 140 °C. However, the acacia bio-oil was a lot better and comparable to HFO in terms of HHV as indicated in Table 4. The acacia oil HHV of 36.809 MJ/kg was higher than the maximum obtained by previous research (31 MJ/kg), as presented in the recent review on fractional condenser systems [24]. Since our focus in this research was on energy derivation from the bio-oil, it is crucial to obtain a larger yield of the higher quality (heavy) bio-oil while considering the extraction of special chemicals from the by-product secondary condensates. This may require fast pyrolysis conditions with a continuous low-moderate flow of purge gas to increase overall yields of the oil to about 75% which, typically should give a higher quality liquid product [20,23]. With the help of fractional condensation, it should be possible to obtain a higher yield of the heavier, high HHV oil, even if it means lowering the condenser temperature to get a good trade-off between the yield and the fuel value of the oil.

Pine bio-oil appeared to be more sensitive to the higher condenser temperature (140 °C), producing low heavy oil yields of inferior quality above the 110 °C condensation temperature, probably due to the presence of molecular compounds with low dew points in the vapor [24]. A sustained high condenser temperature could also possibly cause re-evaporation and recondensation cycles, which could break down the molecular compounds associated with pine bio-oil [20]. The low HHV of the pine-bio-oil at 125 °C could also be attributed to a simple back mixing and pressure drop of vapors within the constricted tubing causing the condensation of an oil-water mixture, which would then trickle back into the primary condenser. Papari and Hawboldt [24] mentioned the pyrolysis of pinewood in a fluidized bed reactor where the primary condenser temperature was varied from 20 to 115 °C while maintaining the secondary one at 20 °C. The yield of dry bio-oil was 20%–55% with 2.5% water and a maximum HHV of 24 MJ/kg. This shows the potential of fast pyrolysis conditions on the yield of dry/heavier oil, which could also be achieved with the fixed bed reactor. The HHV of the pine bio-oil in [24] was higher than that obtained in this research, probably due to the different pyrolysis conditions and regimes used that contribute to bio-oils of various compositions [45].

One possible reason for the highly aqueous secondary condensates could be the inherent, bound moisture in the biomass that is not removed by simple sun-drying. References such as [14,19,30] and recommend oven drying at temperatures between 80 and 105 °C for periods of 24–48 h to remove such moisture and give a higher quality bio-oil. The overall economics of the whole process would need to be assessed for such an energy-intensive pre-treatment step. Indeed, the energy uptake of the pyrolysis process itself has already raised questions on economic feasibility, especially if the product is a low-grade fuel such as bio-oil. However, [48] discussed options of using various energy source models like microwave, renewable solar or parabolic solar concentrators. Reference [7] recommended solar parabolic heating as the most economically attractive and sustainable heating source. Autonomous and semi-autonomous heating designs, where the energy from charcoal and waste gas can be used to provide the energy for the process have also been discussed and experimented on [15,42]. The char obtained, according to [42], contains 25% of the initial energy in the feedstocks, while the pyrolysis process typically requires 15% and the gas can supply 5%. The char from pine, particularly had a higher average HHV of 37.953 MJ/kg compared to that for acacia, with only 29.717 MJ/kg. This implies that the pine dust char would offer a better opportunity for the recovery of energy for the process compared to acacia, per unit weight. Alternatively, the chars can be briquetted and sold commercially to defray expenses.

## 5. Conclusions and Recommendations

In this research, we introduced *Acacia tortilis*, a previously uncharacterized species within the field of pyrolysis, to evaluate its bioenergy potential in addition to reducing the socio-economic impact that this encroacher bush has in Botswana. We compared it with pine sawdust, a waste menace in the neighboring country (Zimbabwe), which is logical since it is one of the most researched feedstocks in pyrolysis. Although fractional condensation has been researched on before, we particularly used a customized lab-scale fixed-bed reactor design for the pyrolysis with the specified cascading condenser types at different temperatures. We observed, from the experiments that increasing the primary condenser temperature improved the calorific value of the acacia heavier oil significantly while reducing the pH and increasing the SG, generally. We concluded that a higher temperature of around 140 °C for the acacia yielded the best quality of oil in terms of the HHV, achieving a value comparable to fuel oil. Such a high temperature also slightly reduced the yield of the heavy oil. We recommend further tests to evaluate the water content, atomization, flash and pour point of both the acacia and pine dust oils and give a fuller picture of the compatibility of the product oil with fuel oil engines. We could not conduct these tests in this research due to the unavailability of the testing equipment. We are, however, concerned that the acacia heavy oil yields are too low with the current intermediate pyrolysis conditions in the fixed bed reactor. In future research, we will switch to the fast pyrolysis mode by using moderate, controlled flow rates of the inert gas in the fixed bed reactor or an alternative fast pyrolysis technology like the fluidized bed. This is because fast pyrolysis is already renowned for its higher yields of better quality oils; therefore selective condensation should enable a larger fractional yield of high-quality oil. It would be ideal to conduct a techno-economic evaluation for alternative process routes, equipment and energy supply and subsequently build a pilot pyrolysis plant for the attractive route, for demonstration purposes. Test runs can subsequently be done using a fuel oil generator for various blends with HFO and, possibly, for bio-oil alone. Our tests indicated that pine-oil already has a low HHV, which disadvantage its use as a furnace or engine fuel compared to the acacia. However, its char has a considerably higher HHV; therefore it can be exploited better through torrefaction to produce smokeless briquettes.

**Author Contributions:** Conceptualization, data curation, writing—original draft preparation, G.C.; writing—review and editing, G.D. and E.M.; supervision, G.D. and E.M. All authors have read and agreed to the published version of the manuscript.

**Funding:** This research received no external funding.

**Acknowledgments:** We acknowledge the support of the Botswana International University of Science and Technology and the University of Johannesburg. We also acknowledge the University of South Africa for access to their equipment for some of the experiments and tests.

**Conflicts of Interest:** The authors declare no conflict of interest.

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
