# Peer review of "Optimizing Yield and Quality of Bio-Oil: A Comparative Study of Acacia tortilis and Pine Dust"

_processes, doi:10.3390/pr8050551_

Round 1

Reviewer 1 Report

The use of renewable resources as basis for fuels production is of present interest. Thus, biomass pyrolysis is an interesting process with good perspectives for its implantation In this respect the manuscript is a contribution and potentially deserves publication. However, the paper has some weak points and must be revised and polished prior considering its publication.A general recommendation in this study regarding paper findings: The paper is quite centered in the features, interest and applications of heavy oil fraction, however its yield is low. Accordingly the overall reliance of the process depends on the quality of all pyrolysis products. This should be taken into account by the authors in the discussion of the results.

The authors should revise citation style in line 40 “Charis et al. (2018) review the research and development…”.

“1.2. Technical background” section should be improved. The authors are encouraged to include a brief discussion of the features and recent application of main biomass pyrolysis technologies, thus, at least the following ones should be reported: fluidized beds, fixed beds, spouted beds, ablative reactors and screw kilns.

Another recommendation in the introduction section is to remark the aim, novelty and relevance of the present paper, preferably, at the end of 1.3 section.

Considering the size of the experimental unit used in this study with biomass batches of 220 g, the denomination pilot plant (line 125) is not actually suitable. The reviewer considers that bench scale plant could be more appropriate.

The selection of experimental conditions, especially pyrolysis and condensation temperatures should be briefly justified.

The description of pyrolysis unit and its operation should be explained in further detail. Thus, reactor dimensions of the detailed design of the condensers should be commented. Are double tube condensers? Was an inert gas flow used in the experiments? What was the flow rate?

Some unexpected results are shown in Table 1. The authors should note that both biomasses showed similar HHVs, however pine dust has higher carbon H and C contents and lower of ashes. Please clarify this point.

The role played by pyrolysis temperature in product distribution is one of the most relevant results obtained in this study. However, it was poorly discussed and must be improved. In addition a comparison with the results obtained in different pyrolysis technologies (see the comment regarding the introduction) could also improve this section.

In Figure 4 and 5 the some results corresponding to the bio-oils condensed at 140 ºC are missing, why? The same happens in Table 3, please clarify the reason of several unavailable results.

Moreover in Table 4 there are some tricky results, as the extremely low HHV corresponding to the bio-oil derived from pine dust condensed at 125 ºC, on the other hand there also extraordinary high HHV (as those corresponding to tortilis at high condensation temperatures), these results are not consistent with reported bio-oil composition, are these values correct?

The percentage of area identified in Table 4 seem to be really low, why?. Moreover the bio-oil composition is mainly includes phenols and methoxyphenol, this result is not in good agreement with other bio-oil analyses from the literature.

Reviewer 2 Report

Abstract (Lines 14-16) The statement should be written, “ Acacia also had higher averages for pH and specific gravity of 3.42 and 1.09, respectively compared to those of pine that were at 2.50 and 1.00, respectively.”

1.2 Technical Background (Lines 84-86) This sentence is incomplete.

2.1 Pyrolysis (Line 121) What is intermediate to fast pyrolysis (IFP)? Can you provide more information?

2.3 Characterization of Products (Line 172) What does m.f stand for?

2.3 Characterization of Products (Line 180) Please include the GCMS method used or provide reference to the method used in the literature in it has been previously published.

4.0 Overall discussion and conclusion (Lines 307-308) Please check the percent heavy oil values as they do not match what is presented in Table 2.

The authors should see and perhaps include the references below:

Mohan, D., Pittman Jr., C.U. and Steele, P.H. Pyrolysis of wood/biomass for bio-oil: A critical review, Energy and Fuels 2006, 20 (3), 848-889.

Oasmaa, A., Solantausta, Y., Arpiainen, V., Kuoppala, E., Sipilä, K. Fast pyrolysis bio-oils from wood and agricultural residues, Energy and Fuels 2010, 24 (2), 1380-1388.

Di Blasi, C., Signorelli, G., Di Russo, C., Rea, G. Product distribution from pyrolysis of wood and agricultural residues, Industrial and Engineering Chemistry Research 1999, 38(6), 2216-2224.

Author Response

I did the corrections for your comments on the very first revision where the editor had asked me to align certain things, therefore they may not be visible in the current 'Tracked changes' document. 

Reviewer 3 Report

The paper in Title "Optimizing yields and quality of recovered bio-oil using process and condenser temperatures: A comparative study of acacia tortilis and pinedust " has a high quality. The topic is interesting and attractive, and the method used during the experimental process is reliable. I suggest a minor revision of its publication. The detailed suggestions are :

The last sentence of the abstract can be canceled. This paper does not include a techno-economic analysis. This sentence is not the major content of this research. Please adding 1.4 in the introduction part to emphasize the major work, general method, and innovation of this paper. 

Author Response

The 'CLEAN' document includes other small grammatical changes that had not been captured by the 'TRACKED'

Round 2

Reviewer 1 Report

The authors have clarified the majority of the questions raised by the reviewer and the clarity and quality of the paper was clearly improved. However, some uncertainties remain regarding the discussion on bio-oils heating values and composition. This point should be further clarified prior considering paper publications.

The authors should note that the composition and features of the bio-oils obtained are far from those usually reported in the literature. Thus, the answer to comments 11 and 12 should be improved. The authors are encouraged to include further comparison with the literature and justify their findings. In fact, the pyrolysis conditions used in the present study do not clearly justify a bio-oil composition with high concentration of phenols and hydrocarbons.
